# An allele-resolved nanopore-guided tour of the human placental methylome

Michaela Kindlova[1,2], Hannah Byrne [1,2], Jade M. Kubler[1], Sarah E. Steane [1], Jessica M. Whyte[1], Danielle Borg [1], Vicki L. Clifton [1] & Adam D. Ewing [1] ✉

The placenta is a temporary organ present during pregnancy that is responsible for coordinating all aspects of pregnancy between the mother and fetus. It has a distinct epigenetic, transcriptomic, and mutational landscape with low levels of methylation, high numbers of transcribed loci, and a high mutational burden relative to somatic tissues. We present this landscape through the application of nanopore sequencing technology to provide a more comprehensive picture of female placental genomics and methylomics along with integrated haplotype-resolved transcriptomic analyses across eight trios. Whole genome sequencing of trios allows robust phasing, permitting comprehensive genome-wide investigation of parent-of-origin methylation and transcription. This enhanced view facilitates identifications of many differentially methylated regions (DMRs), both conserved and differing between individuals, as well as previously unreported imprinted genes including ILDR2 and RASA1 which are potentially important for healthy placental and fetal development.

The placenta is a unique cohort of cell types that collectively present an unusual epigenomic and transcriptomic profile, one that is required for establishing and maintaining healthy foetal development in placental mammals. It has been observed that the placenta has a lower level of DNA methylation when compared to other somatic tissues[1,2], with hypomethylation concentrated in partially methylated domains[3,4]. The placental transcriptome includes a greater number of genes that are uniquely expressed[5] and uniquely depleted[6] than other tissues, perhaps in part owing to the distinctive epigenomic landscape. This landscape includes many imprinted genes, which are expressed predominantly from either the maternal or paternal allele and are a feature of organisms that transfer nutrients directly to their developing offspring[7]. The placenta mediates this interface in mammals and so is a natural epicentre for imprinted genes, where the balance between nutrient consumption and foetal growth is ultimately controlled[8–10].

Here, we take advantage of technological and methodological advancements in nanopore sequencing to better understand the unique epigenomic landscape of the human placental genome. Nanopore sequencing was initially conceived in 1989[11] and has conceptual origins in patch clamp technology developed in the late 1970s for electrophysiology[12]. In recent years, the technology has been subject to rapid development and commercialisation by Oxford Nanopore Technologies Ltd. (ONT). ONT devices offer arbitrarily long and increasingly accurate reads that directly encode both canonical and modified nucleic acid base calls. Through the incorporation of genetic variation, nanopore sequence data can readily be phased into its constituent alleles via read-backed phasing of variants followed by tagging of individual reads[13]. In combination with one of the available highly accurate methods for discriminating between 5mC and canonical cytosine[14], phased long reads enable examination of differentially methylated regions (DMRs) with unprecedented fidelity[15–17].

We carried out whole-genome ONT at greater than 20-fold coverage with a median per-sample N50 of 34kbp, along with Illumina WGS at 30-fold coverage on eight female placentas (Supplementary Data 1). We opted to focus this study on female placentas as they are typically diploid throughout the genome (i.e. two copies of the X chromosome). Previous studies suggest the female placenta induces a greater genomic response to adverse environmental impacts[18]. All

[1]Mater Research Institute, University of Queensland, Brisbane, QLD, Australia. [2]These authors contributed equally: Michaela Kindlova, Hannah Byrne. ✉e-mail: adam.ewing@mater.uq.edu.au

placentas were obtained from participants in the Queensland Family Cohort[19] with uncomplicated pregnancies. Human research ethics approval for the study was granted by the Mater Misericordiae Limited HREC (HREC/MML/73929). Parental DNA was obtained from PBMCs and sequenced on the Illumina NovaSeq 6000 platform to a minimum 30-fold coverage. RNA sequencing was carried out via Illumina sequencing on the same eight placentas with a target throughput of 100 M paired-end reads per sample. Sequencing metrics are presented as Supplementary Data 1.

## Results

### Comprehensive allele-resolved methylation profiling

Backed by long-read genome sequence data, we used informative germline variants to phase the placental genome into maternal and paternal alleles, enabling downstream studies of allele-specific methylation and transcription. This also enables a direct assessment of maternal cellularity in the bulk homogenised placental sample through examination of the variant allele fraction (VAF) of maternally inherited alleles, as substantial deviation upwards from a mean VAF of 0.5 would indicate excess maternal cells in the sample. This was not observed to a notable degree, with all samples showing less than 1% deviation from expectation in VAF for maternally or paternally inherited alleles (Supplementary Data 2), indicating little to no anticipated impact of maternal cells on subsequent analyses.

Comparison of methylation profiles derived from direct modified basecalling of nanopore sequence data enables both a comprehensive genome-wide view of placental methylation patterns and a highly detailed view at individual loci[20]. The placental genome is substantially demethylated relative to the heart, liver, and hippocampus tissues we have previously analysed[21], with an apparently bimodal distribution of methylation levels (Fig. 1a). Visualisation of the chromosome-wide methylation pattern shows that, with the exception of the X chromosome, relative demethylation is not chromosome-wide, but is concentrated in distinct regions (Fig. 1b, see Supplementary Fig. 1 for all chromosomes). This is consistent with the prior observation of partially methylated domains (PMDs) across the placental methylome[3,4]. To provide orthogonal verification of

methylation profiles via another method, we carried out EM-seq[22] on two samples (placentas 054 and 103), yielding highly concordant methylation patterns (Supplementary Fig. 2, mean per-chromosome Pearson's r 95% c.i. = 0.98 ± 0.01).

Using read-backed phasing[13] coupled with parental variants, attribution of reads as maternal or paternal in origin is straightforward. Comparison of alleles on a chromosomal scale yields highly similar methylation profiles between maternal and paternal autosomes (Fig. 1c, Supplementary Fig. 3). The X chromosome is an interesting exception, with some but not all placentas showing globally higher methylation of either the maternal or paternal chromosome. (Fig. 1c, Supplementary Fig. 4) We ascribe this to clonal X-inactivation, the dosage compensation mechanism whereby one of the two X chromosomes is transcriptionally repressed in development. Clonal expansion of placental tissue with one or the other X inactivated yields a skewed geographic profile of maternal or paternal-specific methylation across the X chromosome, consistent with allele-specific methylation of the XIST promoter (Supplementary Fig. 5), and as previously described[23]. The use of bulk tissue captures a stochastic clonal profile when homogenised, yielding the observed profiles.

On a locus-specific level, we observe many known differentially methylated regions (DMRs) such as those associated with IGF2/H19, and GNAS (Supplementary Material). Extending this to catalogue all DMRs, we identify 723 DMRs in total, with 184 specific to this study and 539 previously identified DMRs as annotated by a collection of studies providing specific DMR coordinates[24–29] (Supplementary Fig. 6, Supplementary Data 3). Owing to diminished coverage of heterozygous sites in short-read data, comparatively few of these allele-specific patterns were reliably observable via short-read methods, with EM-seq data covering 155 sites in both samples that were subjected to EM-seq (Supplementary Data 4). Of these, 146 of 155 sites (94%) showed the expected DMR, showing high concordance where technically possible. Of the 184 newly identified DMRs, 41 are near genes annotated as at least putatively imprinted, while 143 are not near a known imprinted gene. DMRs were observed to include those consistent across all samples and those present in fewer than all eight placentas analysed, consistent with polymorphic DMRs noted in previous studies[25,29,30].

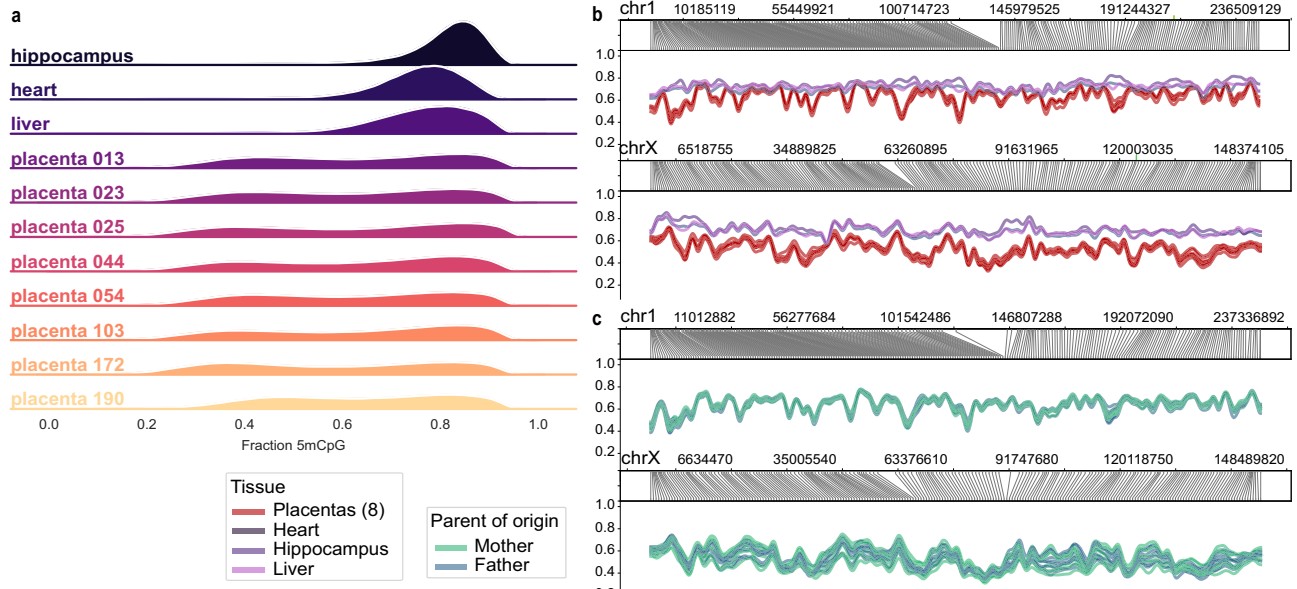

**Fig. 1 | Large-scale methylation patterns in human placentas. a** The distribution of methylation levels (fraction of CpGs with 5mC in 10kbp bins) for non-placental tissues versus placentas in this study. **b** The same samples as in (**a**) showing the distribution of methylation across chromosomes 1 and X. For all chromosomes see

Supplementary Fig. 1. **c** The methylation profiles for placental samples phased into paternal and maternal profiles are shown for chromosomes 1 and X. For all chromosomes see Supplementary Fig. 3.

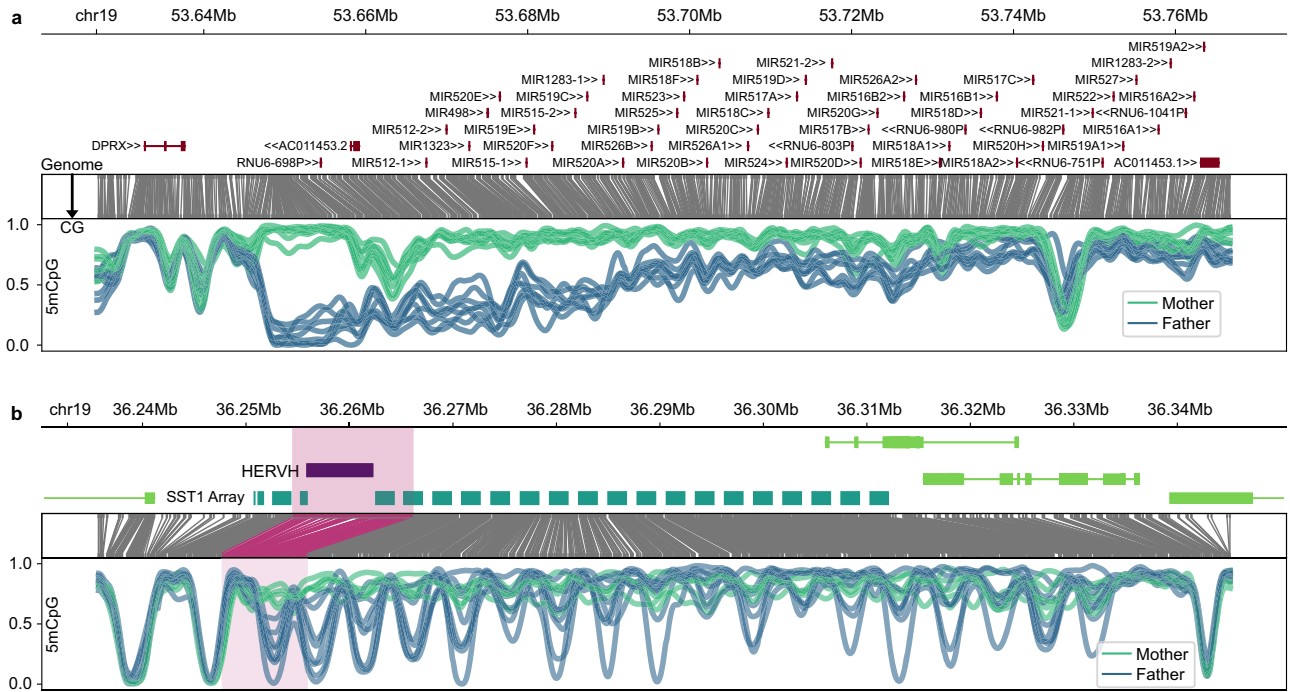

**Fig. 2 | Examples of regions with notable allele-specific methylation patterns in placenta.** From the top down, each panel shows the genomic coordinates, relevant annotations, translation from genomic space into CpG-only space, and a smoothed methylation plot at the bottom. The C19MC locus (**a**) contains many paternally-expressed (Supplementary Fig. 7) and paternally-demethylated pri-miRNA loci and is among the best known examples of a placental DMR. Here we can see the pattern of differential methylation across the locus. The SST1/MER22/NBL2 macrosatellite array (**b**), also on chromosome 19, is a previously unreported example with a periodic pattern of DMRs corresponding to the SST1 repeats, originating at a HERVH element upstream of the array. This locus is also paternally expressed, as shown in Supplementary Fig. 8.

Plots for all DMRs are available as Supplementary Data at https://doi.org/10.6084/m9.figshare.22589635.v1.

A known visually striking example is the chromosome 19 micro-RNA cluster (C19MC), which encodes approximately 50 paternally-expressed miRNAs in the placenta[31]. The miRNAs at this locus may be essential for trophoblast differentiation[32], and their deregulation may be linked to pre-eclampsia and other complications[33]. The extent and the consistency of the differential methylation, as shown in Fig. 2a, are remarkable, and we observe a pattern of steadily decreasing differential methylation across the locus.

To highlight an interesting example of a previously uncharacterised DMR-rich region, we identified that an array of SST1 macrosatellite repeats on chromosome 19 has a curious methylation pattern, which exhibits periodic methylation and demethylation of the paternal allele corresponding to the positions of SST1 repeats (Fig. 2b). The locus is also paternally expressed with transcription initiating at an intact HERVH retrotransposon upstream of the differentially methylated portion of the array (Fig. 2b). While SST1, also known as MER22 or NBL2, is highly variable in copy number and is present in arrays at multiple locations in the genome, this particular array on chr19q is one of the more ancestral and stable instances[34]. This finding is also potentially consistent with HERVs being well-documented drivers of transcriptional innovation in the human placenta[35].

Finally, our data confirm a strong bias toward imprinted paternal demethylation in the placenta[36] (Fig. 3a), with the large majority of DMRs being demethylated on the paternal allele relative to the maternal allele. Comparison of parent-of-origin between placenta and the NA19240 lymphoblastoid cell line[37,38] shows that this bias is not present in NA19240 (Fig. 3a). When compared to other tissues and cell lines, most placental DMRs show a greater absolute magnitude of differential methylation in placentas when compared to the same site in other tissues or cells (Fig. 3b). With the caveat that a limited number of nanopore-sequenced tissue samples are available for comparison at this time, this indicates a high degree of placental specificity for the DMRs identified here, along with some cross-tissue conservation of DMRs, which is also of interest.

## Allele-resolved expression

Deep cDNA sequencing (RNA-seq) was carried out on the placental samples. Compositional inference of cell types contributing to bulk gene expression profiles via CIBERSORTx[39] using single-cell sequence data from a prior study[40] indicated a comparable mix of cell types across samples (Supplementary Fig. 9). Short RNA-seq reads were separated into maternal and paternal components where possible through the presence of one or more phased variants. Multidimensional scaling of the overall transcriptional profile suggests the variance is largely between individuals rather than between parents of origin, as one might expect (Fig. 4a). In contrast, when this analysis is carried out on loci immediately surrounding DMRs, parent of origin emerges as a substantial source of variance, reflected in the separation of maternal and paternal expression (Fig. 4b). As phasing is limited to reads that contain informative variants, the ability to phase short-read RNA-seq data is limited, particularly for RNA-seq where coverage is biased towards exons. Furthermore, estimates of differential allelic expression over a given locus may be possible for some individuals but not others, given that informative variants are segregating. Nonetheless, imprinted genes are readily identifiable through differential gene expression between maternal and paternal alleles; an analysis of allelic differential expression of genes identifies 74 imprinted genes at FDR < 0.25 (Fig. 4c and Supplementary Data 5).

## Discovery of previously unreported imprinted genes

Intersecting the DMRs described here with our allele-specific expression results has the potential to uncover previously unrecognised imprinted genes. The most prominently notable examples are ILDR2,

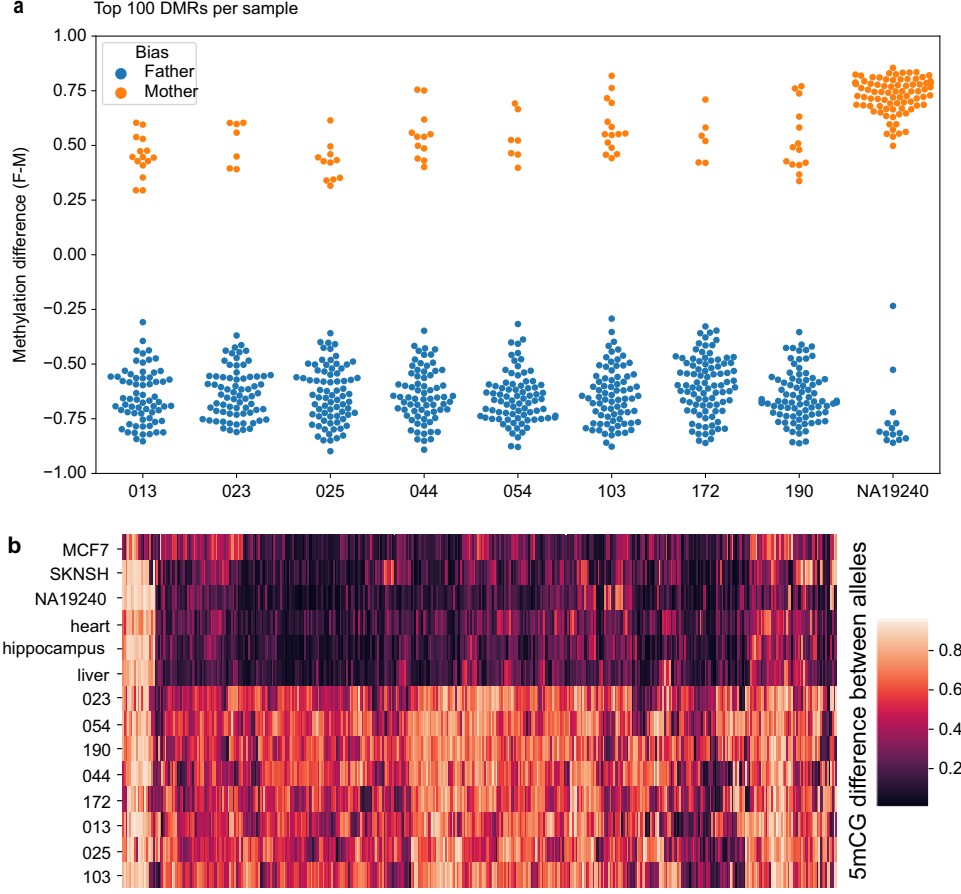

**Fig. 3 | Properties of placental DMRs. a** Placental DMRs are highly biased towards paternal demethylation when compared to a non-placental sample (NA19240, EBV-transformed lymphoblastoid cell line). Methylation difference in (**a**) is defined as paternal minus maternal methylation fraction. For each sample, the top 100 DMRs based on our analysis are shown. **b** Placental DMRs are more differentially methylated in placenta versus other tissues or cell lines. Lighter colour corresponds to a greater absolute difference in methylation fraction between alleles.

which appears to be paternally demethylated and expressed, and RASA1, which is maternally demethylated and expressed.

ILDR2 is a member of the Ig superfamily and is a B7-like protein known to modulate T cell activity, thereby potentially serving an important function in the regulation of autoimmune responses[41,42]. To our knowledge, ILDR2 has not previously been identified as imprinted. Our data indicate a strong paternally-demethylated DMR corresponding to ENCODE-derived CTCF-bound cis-regulatory elements around the penultimate exon of most ILDR2 isoforms, with essentially all transcription derived from the paternal allele (Fig. 5a). Together, this suggests ILDR2 is a previously unreported paternally-expressed imprinted gene that may play a role in suppression of active T cells in the human placenta.

RASA1 (RAS p21 activator 1 or p120-RasGAP) is a previously unreported example of a maternally-expressed imprinted gene (Fig. 5b). It is part of the RAS/MAPK pathway, where studies in trophoblasts suggest it plays an inhibitory role, and its expression may attenuate trophoblast proliferation and invasion in the placenta[43]. The maternally expressed imprinted gene MEG3 has been suggested as an upstream silencer of RASA1 in this context via recruiting of EZH2, leading to H3K27me3-mediated silencing of RASA1[44]. Beyond the placenta, mutations impacting RASA1 are implicated in an autosomal dominant disorder with variable presentation, including abnormal vascular growth and limb hypertrophy[45,46]. The maternally-demethylated imprinted DMR is downstream of a nearby paternally-demethylated imprinted DMR, and seems to correspond to imprinted expression of the lncRNA LINC01949 (Fig. 5b). Together, the region

encompassing the 5′ end and upstream of RASA1 appears to represent an imprinting control region for multiple genes exhibiting both paternal and maternal imprinting, as has been observed at other ICRs.

Tumour suppressor candidate 3 (TUSC3) is an example of "epipolymorphism", where a previous study remarked on the variability of maternal-specific methylation and a potential association with preeclampsia[30]. Here, we can confirm the specific location of the DMR at the TUSC3 5′UTR as well as show a corresponding bias towards expression of the paternal allele (Supplementary Fig. 10). It is interesting to note that gene body methylation of TUSC3 has an inverse relationship with DMR methylation in that while the paternal allele is demethylated relative to the maternal allele in 4 of 8 samples, the opposite is true of the gene body. This is consistent with a high level of methylation of transcribed gene bodies previously observed in the placental methylome[4], as elsewhere. WNT2 is another epipolymorphic gene noted in the same study[30] with a comparable pattern of inconsistent differential methylation with differential parent-of-origin expression (Supplementary Fig. 11).

### Genome variation

It has been noted that an unusually high number of somatic mutations are detectable from bulk placenta[47,48]. Comparison of placental genomes to parental genomes indicated an average of 186 point mutations (single base substitutions and short insertions/deletions) per sample not detectable in the parental genome or in gnomAD at an allele frequency greater than $10^{-5}$ (Fig. 6a and Supplementary Data 6). On average, 102 of these have a VAF significantly below 0.5 as assessed by

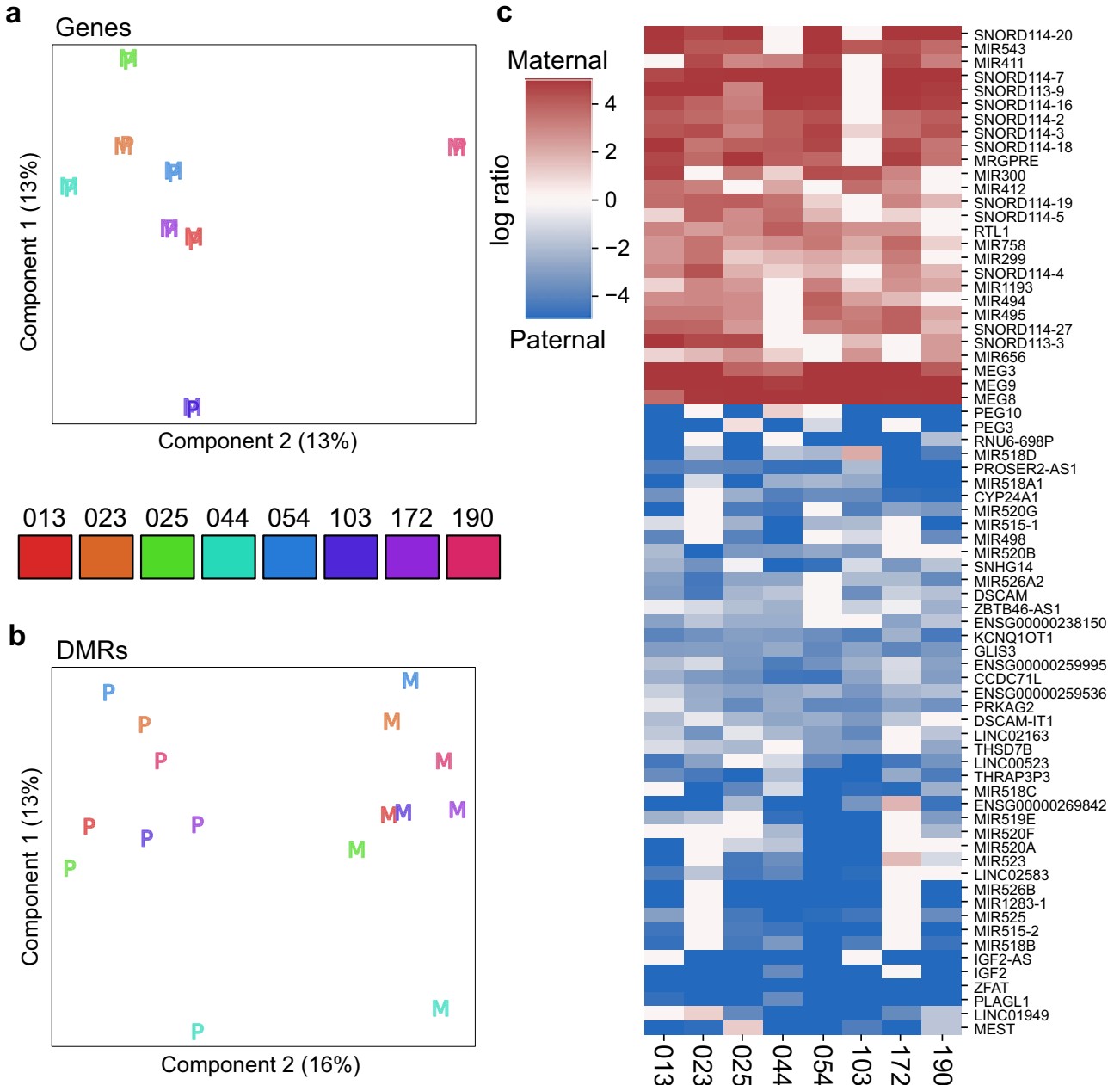

**Fig. 4 | Allele-specific expression via read-backed phasing.** Overall (**a**), multi-dimensional scaling shows variability in gene expression is attributable to differences between individuals with negligible variance due to maternal versus paternal allele. This is the expected result given most expressed loci are not imprinted or expressed in an allele-specific manner. When focusing specifically on expression surrounding DMRs (±5kbp) (**b**), parent-of-origin becomes a major driver of variance in gene expression at DMR loci. Statistical analysis of allele-specific expression (**c**) shows many known imprinted loci - most of the maternally expressed examples are non-coding and short RNAs clustered around the MEG3/8/9 locus on chromosome 14. The paternally-expressed loci are more heterogeneous but show a plurality of paternal expressed miRNAs at the C19MC locus (Fig. 2a). As genetic variability involving informative phased SNVs induces considerable dropout (white cells in heatmap), and consequently high variability in expression, the FDR is set at 0.25 to capture the most differentially expressed loci for this figure.

a one-sided binomial test; a VAF lower than 0.5 suggests a de novo somatic mutation rather than a de novo constitutive mutation. This finding is consistent with a recent report of widespread somatic mutation in placental genomes[47].

Placenta-specific structural variants were considerably less numerous, with only 4 identified concordantly in both short and long read data (Supplementary Data 6b). The most notable among these is a 44.7 kbp duplication involving most of CNDP1 and the promoter of a downstream gene, ZNF407 (Fig. 6b). While CNDP1 is not expressed in the placentas without this duplication, the sample with the duplication expresses CNDP1 downstream of the duplication 5′ junction. The

promoter of ZNF407, which is highly expressed in all eight placentas, presumably drives transcription of CDCP1 in the sample with this structural rearrangement. Based on VAF (~0.1 for both Illumina and Nanopore), absence from the parental genomes, and a relatively low level of ZNF407-driven CNDP1 expression as compared to ZNF407, it seems likely that this duplication is somatic. While there are examples of placenta-specific duplications associated with adverse foetal outcomes[49], that was not the case with the duplication reported here. However, this does add to evidence that a small minority of somatically acquired mutations affect the placental transcriptome, and a subset of these instances may impact foetal development.

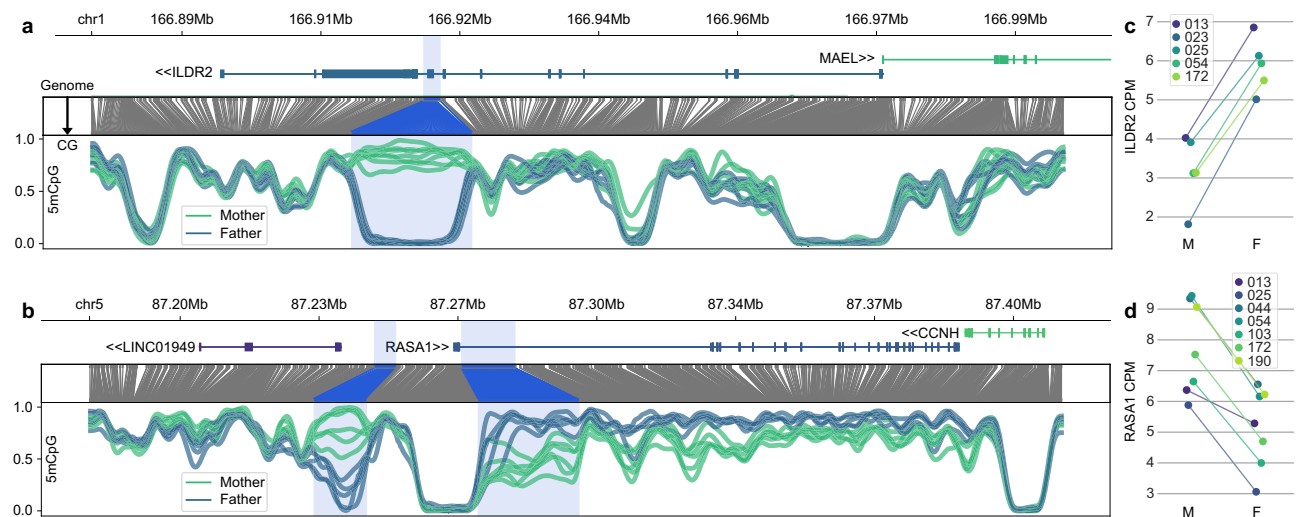

**Fig. 5 | Previously unreported imprinted genes. a** ILDR and **b** RASA1. Here, ILDR2 is paternally expressed (**c**) while RASA1 is maternally expressed (**d**). The lncRNA upstream of RASA1 (LINC01949) appears to be paternally expressed. The correspondence between genomic coordinate space and CG-only space is shown below the gene annotations, followed by smoothed CpG methylation fraction. Highlights correspond to DMR locations. CPM counts per million, M mother, F father.

**Fig. 6 | Detection of de novo placental variants. a** Variant allele fractions (VAF) of point mutations identified from Illumina sequencing in one placenta only, absent from all parental samples, and confirmed in Nanopore sequence data. VAFs significantly below 0.5 via proportion test using read counts are highlighted in dark red. **b** Identification of a duplication involving most of CNDP1 and the promoter of ZNF407 (red box). RNA-seq read depth is shown for the sample with the duplication (023, orange) and a representative sample without the duplication (013, pink). **c** Elevated CDNP1 expression (CPM counts per million) is unique to the duplicated sample and may be due to the rearrangement bringing the promoter of ZNF407, which is expressed in all 8 placentas, upstream of the duplicated portion of CDNP, consistent with the read depth plot.

## Discussion

In this study, we have applied the advantages of nanopore sequencing technology to the unusual epigenomic environments of the human placenta, yielding insights and avenues for further study. Through the application of this technology to the samples available through the Queensland Family Cohort, we have been able to explore the allele-specific methylation profiles of a placental cohort to identify a substantial number of DMRs not previously described. We have confirmed and provided better visualisation of hundreds of known DMRs alongside identification of many previously unreported DMRs and imprinted genes. Overall, differential methylation between parental alleles is more prevalent than differential gene expression between alleles. Speculatively, this could indicate loci primed for differential gene expression between parental alleles, but perhaps occurring in a different cell type than the placenta or at an earlier developmental time point.

The genetic conflict hypothesis is a leading theoretical explanation for genomic imprinting that posits a motive for regulation of genes, particularly those involved in growth, in a manner specific to either the maternal or paternal allele[7]. In this paradigm, the regulation of the maternal genome has evolved to optimise the distribution of resources between her and her offspring in a way that is compatible with multiple offspring over time. This conflicts with the paternal genome, which has evolved a regulatory regime to support as much growth as possible in each offspring. Therefore, the argument follows that genes driving placental growth will tend to be paternally imprinted while genes that inhibit growth will tend to be maternally imprinted, as reviewed in Wilkins and Haig[50]. The previously unreported imprinted genes we highlight here are compatible with this model. RASA1, which is maternally demethylated and expressed, inhibits the activity of the RAS/MAPK pathway, thereby inhibiting trophoblast proliferation and invasion. The paternally demethylated and expressed ILDR2 modulates activated T cells, which is consistent with promoting placental growth through reduction of maternal immune response towards the placenta.

This study highlights the technical capabilities and utility of nanopore sequencing technology and associated software tools, with their combined ability to capture modified bases (5mCpG in this manuscript) and genomic variation. Short-read data is included here as a point of comparison and as an economical means of obtaining parental genotypes. Short-read data as a source of variants for allele-specific methylation is not generally necessary, given improvements in sample preparation, flow cell technology, and base calling methods coupled with improved variant calling software. What has been uncovered here forms the basis for using a similar approach to study the function of these loci in a targeted manner and the impetus for expanding this cohort such that association between variation in DMR status and relevant outcomes is adequately powered. Male placentas are a notable omission from the present study, and their future inclusion may enable additional insights into why females are better able to adapt to environmental stressors *in utero* than males[18]. The unique epigenomic profile of the placenta surely contains many yet unrecognised signals linked to the developmental origins of health and disease.

## Methods

### Inclusion and ethics

Our study design and conduct are compliant with all relevant regulations regarding the use of human study participants as set forth in the Australian National Health and Medical Research Council (NHMRC) National Statement on Ethical Conduct in Human Research (Updated 2023). This study was conducted in accordance with the criteria set by the Declaration of Helsinki. All placentas were obtained from participants in the Queensland Family Cohort[19] from uncomplicated pregnancies. Written informed consent was received from all participants

(mothers and partners). Participants were not compensated for participation in this study. Human research ethics approval for the study was granted by the Mater Misericordiae Limited HREC (HREC/MML/73929).

### DNA sequencing

Female (assigned at birth) placental samples and parental blood samples from uncomplicated pregnancies with no significant placental abnormalities delivered vaginally at term were obtained from the Queensland Family Cohort (QFC)[19]. Selected statistics regarding placentas, neonates, and parents are presented in Supplementary Data 7. The QFC follows the placental collection and processing protocol of the Global Pregnancy CoLab[51], utilising multi-site sampling of chorionic (foetal) villous tissue (minimum 4 sites per placenta). Samples were washed to minimise carry-over of maternal blood and pooled prior to being snap-frozen. High molecular weight (HMW) DNA for sequencing was extracted from the placental tissue using the Nanobind Tissue Big DNA kit (NB 900-701-01, Circulomics). For each placenta, approximately 40 mg of tissue was homogenised using a pre-sterilised pestle in lysis buffer at 4 °C, and DNA was extracted as per manufacturer's protocol. Where required, samples were additionally purified with Ampure Beads XP (A63881, Beckman Coulter) and 80% EtOH. Samples were treated with the SRE kit (SS-100-101-01, PacBio) as per manufacturer's protocol to eliminate fragments <25 kb prior to sequencing.

DNA for short-read whole-genome sequencing was extracted using a DNEasy kit (69504, Qiagen). For the placental samples, approximately 35 mg of tissue was used as the starting input, and DNA was extracted as per manufacturer's protocol with a tissue lysis time of 90 min. For the parental genomes, 250 μl of PBMCs in 10% DMSO was washed twice with PBS and then resuspended in 250 μl of PBS before DNA was extracted as per the manufacturer's protocol. Concentration and purity of all DNA samples were assessed by Qubit and nanodrop, respectively.

Short read whole genome sequencing was carried out on all samples by Macrogen Oceania on an Illumina NovaSeq 6000. Short reads were mapped to the reference genome (hg38, GATK Resource Bundle) using bwa-mem2[52] 2.0pre2, and duplicates were marked via samblaster[53] 0.1.26.

Long-read nanopore sequencing was carried out on the placenta samples at the Australian Centre for Ecogenomics on an Oxford Nanopore Technologies (ONT) PromethION on version 9.4.1 flow cells. Where required to obtain comparable coverage across all samples, additional top-up sequencing was carried out on ONT Mk1c devices using version 9.4.1 flow cells. Bases (A, C, G, T, 5mCpG) were called with Megalodon 2.5.0 using the models included with guppy 5.0.7 dna_r9.4.1_450bps_modbases_5mc_hac_prom for Promethion data and dna_r9.4.1_450bps_modbases_5mc_hac for MinION data. Reads were aligned to the reference genome (hg38, GATK Resource Bundle) using minimap2[54] 2.22. Quantification of methylation and visualisations were created using methylartist[55] 1.0.6.

Enzymatic methylation sequencing (EM-seq)[22] was carried out according to the manufacturer's protocol (NEBNext Enzymatic Methyl-seq Kit E7120, New England Biolabs). Two barcoded NEBNext libraries were sequenced on one lane of an Illumina NovaSeq X by Azenta Life Sciences. Sequence data was quality controlled and processed using BISCUIT[56] 1.4.0. The resulting. bam files with "MD" tags were processed via "methylartist db-sub", which converts C-T substitution data into a format readable by methylartist for direct comparison to nanopore-based methylation data.

### cDNA sequencing

Total RNA was isolated from the frozen placenta samples using an RNeasy extraction kit (74104, Qiagen). Placental tissue was kept frozen by liquid nitrogen and powdered using mortar and pestle. Aliquots of approximately 30 mg of tissue were resuspended in a lysis buffer

containing 1% $\beta$-mercaptoethanol and immediately homogenised using ceramic beads (6540434, MP Biomedicals) on a Bead Ruptor homogeniser (Omni Inc). Samples were subsequently spun at 13,000 × $g$ for 10 min, supernatant was moved to a new tube, and RNA was isolated according to the manufacturer's protocol. All samples were treated with Turbo DNase (AM2238, Invitrogen) to avoid DNA contamination. Purity and concentration of RNA samples were checked on an Agilent Bioanalyser (RIN scores of samples used for further analysis ranged from 5.8 to 7).

Synthesis of cDNA using a RiboZero Plus protocol and sequencing were carried out by the Australian Genome Research Facility (AGRF) on an Illumina NovaSeq 6000 (150 bp PE) with a total target depth of 100 M PE reads per sample. Reads were aligned to the reference genome (hg38, GATK Resource Bundle) using STAR[57] 2.7.10a, and duplicate reads were marked via samblaster[53] 0.1.26. Aligned reads were assessed against quality control metrics using rnaseqc[58] 2.4.2 and CollectRnaSeqMetrics from picard 2.27.5. Reads were phased using whatshap-phased variants as per long-read DNA sequencing. Reads were counted against DMR locations and against Ensembl gene build 106 using featureCounts from Rsubread[59] 4.1.2. Read counts were assessed for differential expression using edgeR[60] 4.1.2.

Assessment of cell type composition was carried out via the CIBERSORTx[39] Docker container. A single-cell sequence data read count matrix was obtained from GEO accession GSE182381[40], and cell fractions were estimated with parameters --single_cell TRUE --replicates 20 --perm 100 --verbose TRUE --rmbatchSmode TRUE.

### Mutation detection

**SNVs/INDELs.** To detect germline variants used for phasing, variants for each trio were called jointly with GATK 4.1.9 HaplotypeCaller, and quality scores were recalibrated using GATK VQSR. Putative de novo and somatic variants were detected using GATK Mutect2[61], with gnomAD[62] 3.1.2 as a germline resource, using the set of all parental genomes as a "panel of normals". Mutect calls were filtered via GetPileupSummaries, CalculateContamination, and FilterMutectCalls and screened using a script to remove variants with a gnomAD allele frequency > 1e$^{-5}$ and total read depths <10. Putative de novo and somatic mutations detected in placentas from the Illumina WGS data were verified in the nanopore data from the same placenta.

**SVs.** Structural variants were detected from short-read data via delly[63] v1.1.6 on a per-trio basis, filtered for somatic variants in the placenta sample, genotyped against the whole cohort, and further filtered for somatic variants versus the whole-cohort genotyping results. SVs from long-read data were detected via sniffles2[64] version 2.0.4 using the −non-germline and −phased options, and delly-derived somatic SVs confirmed via these results were reported. Additional tools were used to screen for transposable element insertions: TEBreak[65] for short-read data and TLDR[21] for long-read data.

### Phasing

Variants called from Illumina, along with pedigree information (trios), were phased using nanopore reads via WhatsHap[13] v1.4. Phased VCFs were used to tag nanopore reads using a combination of WhatsHap and customised methods. Manual examination of known DMRs (e.g. IGF2, GNAS, PEG3) was used as a quality control check.

### DMR detection

Genome-wide phased methylation values at each CpG were output via "methylartist wgmeth --dss", which yields outputs suitable for import to the DSS differential methylation package[66] implemented in R. DSS was used on both a per-individual (e.g. Fig. 3a) and whole-cohort basis (Supplementary Data 3) to identify differentially methylated loci. Candidate DMRs were ranked by "areaStat" (aggregated DML statistic). These were plotted with a ±20kbp window using "methylartist locus"

and visually inspected to filter DMRs. Known DMRs were intersected via "bedtools intersect" to construct a single list of known DMRs, and these were screened against DMRs detected via our approach. Previously reported DMRs that were not identified by our approach were also plotted via "methylartist locus" and visually inspected to assess whether the known DMR was apparent in our cohort.

### Reporting summary

Further information on research design is available in the Nature Portfolio Reporting Summary linked to this article.

## Data availability

All sequence data generated in this study have been deposited at the EGA with accession EGAD50000001850 via controlled (mediated) access due to human research ethics requirements. We endeavour to respond to access requests within two weeks. Processed data is available in the form of allele-specific methylation levels for all placental samples as indexed bedMethyl format at https://doi.org/10.6084/m9.figshare.28593515.v1. Additional Supplementary data is available at https://doi.org/10.6084/m9.figshare.22589635.v1. Nanopore sequence data from Heart, Liver, and Hippocampus tissues[21] was obtained from SRA Bioproject PRJNA629858. Nanopore sequence data from NA19240[37] were obtained via ENA accession PRJEB26791. MCF-7 sequence data[55] was obtained from SRA Bioproject PRJNA748257. Placenta single-cell sequencing read counts[40] were obtained from GEO accession GSE182381.

## Code availability

Primary analyses were conducted via Methylartist[55]. Scripts for using methylartist output in DSS are included in the 'scripts' directory in the methylartist repository (https://github.com/adamewing/methylartist). Specific methylartist commands, additional scripts for generating figures, and supporting data specific to this study are available at https://github.com/adamewing/KindlovaByrne2025 (https://doi.org/10.5281/zenodo.15532400) and at https://doi.org/10.6084/m9.figshare.28636532.v1.

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

## Acknowledgements

We gratefully acknowledge the families participating in the Queensland Family Cohort study, which enabled this study. We acknowledge the Translational Research Institute (TRI) for research space, equipment, and facilities. This study was funded by the Australian Department of Health Medical Research Future Fund (MRFF) (MRF1175457 to A.D.E.), by the Mater Foundation, and by the Advance Queensland Women's Research Assistance Programme (WRAP086-2019RD1 to M.K.). We acknowledge the sequencing service providers used in this study: The Australian Centre for Ecogenomics (ACE), Macrogen Oceania, and the Australian Genomic Research Facility (AGRF). We would like to acknowledge Natasha Jansz for helpful discussions.

## Author contributions

A.D.E. designed the study and analysed sequencing data. A.D.E., M.K., and H.B. wrote the manuscript with feedback and input from all authors. M.K., H.B., J.M.K., S.E.S., and J.M.W. carried out sample preparation and quality control. J.M.K. and S.E.S. provided valuable advice and expertise concerning the preparation of placental RNA. V.L.C. and D.B. facilitated provision of samples on behalf of the Queensland Family Cohort. V.L.C. provided expertise and advice concerning study design and data analysis.

## Competing interests

A.E. has received reimbursement for travel, accommodation, and conference fees to speak at events organised by Oxford Nanopore Technologies. The remaining authors declare no competing interests.
