## [Transparent Peer Review file · Nature Communications]

An allele-resolved nanopore-guided tour of the human placental methylome

Corresponding Author: Dr Adam Ewing

Version 0:

Reviewer comments:

Reviewer #1

(Remarks to the Author)

In this manuscript, Kindlova, Byrne et al. present a phased methylome analysis of human female placentae. The study design is excellent, with trio data and high levels of replication for this type of study (8). The underlying data (high-depth nanopore + Illumina) is excellent as well. The authors confirm the lower methylation levels in placenta relative to most other tissues, call allele-specific DMRs including some new ones (pending comparison with more recent studies), describe 2 new imprinted genes, and detect somatic mutations.

While the analyses are generally well done, the manuscript is mostly illustrative: the approach of using nanopore for phased methylomes is already well described and the analyses are not pushed far enough to test the hypotheses the authors put forward as explanations for the patterns they see. The impact for either the nanopore genomics or the epigenetics field is therefore limited.

Comments:

1. the manuscript should acknowledge at least two precedent studies for nanopore-based phased methylomes: Gigante et al. NAR 2019 (placenta imprinting already, but in mouse) and Akbari et al. eLife 2022 (imprinting in 12 human B-lymphocyte cell lines)
2. The Akbari DMRs should be included in the cross-reference for known DMRs
3. what characterises the DMRs that were previously missed? Do they belong to certain regions that couldn't be phased before, or are they tissue-specific? Were there known DMRs that couldn't be found? Can the DMRs be classified into primary and secondary imprints?
4. Please give more details about how the DMRs were selected. The methods imply that DSS was used to generate candidates, which are then selected visually?
5. Care should be taken with the terminology around "imprinted". For gene expression, just say "paternally expressed" or "maternally expressed". "Paternally imprinted" is ambiguous, usually it means it's the paternal allele that carries the primary imprint, but it says nothing of whether the imprint silences or activates the gene.
6. Easily testable hypotheses should be tested: 1) PMDs contributing to lower placenta methylation, why not segment the PMDs and quantify them? 2) clonal skew in X chromosome inactivation, why not compare the skews to the Xist promoter methylation skew?
7. The read methylation heatmaps are far too cramped, I can't tell where one sample begins and the next starts. Since the manuscript emphasises visualisation, these should be improved. The "expression space" transform is also not intuitive. Lining up this plot with the gene annotation misleads into comparing the two tracks, when it's basically impossible to do so. Traditional read pile-ups over exons would be much easier to understand, or simply a dot plot of mat/pat cpms for each sample. The genomic position markers are also very hard to parse on all methylart plots, so it's hard to tell the scale of the plot. One could add commas (100,000,000 b) or simpler units (100 Mb).

8. NA19240 (Fig 3a) has an opposite bias, with more paternal methylation. Is this expected? Is this observed in other replicates (see Akbari)? For Fig3b clarify that the set of DMRs is the 723 DMRs identified in this study. For Figure 3b, use a divergent scale from -1 to +1 to show that the methylation difference in the other tissues follows the same direction (or not).
9. For Fig4a, colour by sample and add %variance explained to the axes.
10. Is 100 somatic mutations a lot? Give a quantitative comparison with previous estimates for placenta and other tissues.
11. give statistics about %genome or %data haplotyped.
12. There's an opportunity to comment in the discussion on how one would conduct this study today: with R10.4.1 chemistry, there would be no need for Illumina data, and allele-specific expression would be much better quantified with long-read transcriptomics (e.g. PCB111).

Reviewer #2

(Remarks to the Author)

Summary:

The placenta is a developmentally critical and relatively understudied tissue, which is highlighted in the current paper. In a subsample of the Queensland Family Cohort, 8 families were selected for analysis. In each family, maternal and paternal PBMCs were whole genome sequenced using standard, Illumina short read approaches. Placental tissue DNA was measured Nanopore long read sequencing, which allows for both sequence variant calling, as well as modified cytosine calling for DNA methylation assessment. The maternal and paternal genomes were used to phase the placental genome and test for differentially methylated regions between the maternal and paternal genomes. They identified 723 DMRs, including 536 that were consistent with previous reports. They also measured placental gene expression and linked DMRs to differentially expressed genes. Two regions, related to ILDR2 and RASA1 were highlighted as novel and with interesting biology. Finally, they noted somatic mutation burden in the placenta. This manuscript represents a creative use of a cutting-edge technology to understand the placental omics landscape. It provides an important contribution to the field, and I am highly enthusiastic about future opportunities to investigate these newly characterized imprinting regions in other cohorts.

Critique:

The authors begin the introduction by mentioning the unique cohort of cell types in the placenta. Have the authors estimated the cell composition in their samples? Methods to estimate placental cell composition using sorted-cell DNA methylation (PMID: 33407091) and single cell RNA sequencing (PMID: 36914823) profiles are available. I'm curious if the between sample variability in DMRs observed may be reflective of a differential abundance of a specific cell type that is driving the imprinting signature.

While this is a very compelling scientific story, there are elements of the communication that could be adjusted to facilitate interpretation for the reader. For example, many of the figure axis and color bars lack labels, requiring the reader to sort through long legends and make assumptions. The black/white/blue/green fuzzy heatmap in figure 2 and 5 is particularly difficult to interpret as I am not seeing any closed dots at the current resolution, I don't know the meaning of all of the colors, and the number of rows appears to be more than the number of samples.

For future cohorts to compare with this work and assess generalizability, the Methods section would benefit from a section on participant recruitment and sample collection. It was part of a larger cohort, but how were these 8 families specifically selected? When was parental blood drawn (at delivery)? How were placental tissue samples collected and archived? Was maternal blood washed from the placentas?

Methods: For the genetics data, how many base pairs were the long and short read sequences? How many million reads were generated per sample?

Methods: "Genome-wide phased methylation values at each CpG were output via "methylartist wgmeth --dss"... This statement would be useful to have a citation so others could find this method. What significance level and magnitude level were used to call a DMR?

Methods: Especially for your placental DNA methylation readers who more commonly use bisulfite sequencing and are newer to Nanopore, it may be useful to have a sentence in the methods that says how the DNA methylation levels were quantified in this paper.

While I am enthusiastic that the data will be available, I did not see a code availability statement.

Discussion: Can the authors put these DNA methylation observations about X-inactivation and imprinting in the context of developmental timing, relative to their measurement in placental tissue at term? When during development does X-chromosome inactivation take place and is it reasonable to assume that the size placenta pieces measured in this paper all came from the same clonal expansion, or might there be more than one clone represented in these 35mg of tissue?

Similarly, when during development do the two waves of epigenetic wiping take place and how does that relate to this term collection?

I think that Supplemental Table 3 is going to be incredibly useful for other cohorts to examine replication or enrichment of these imprinting sites. Could the authors expand on the column titles or include a table legend to explain all of the information that is included. For example, in the column "Placenta_Mean_DMe", which parent is the reference group? It looks like maternal, since many of the values are negative, but want to be sure. Is there a p-value column? I really like how they point out DMRs that were highlighted in other papers in this table. Were there imprinting regions suggested in those other papers that were not identified here?

Reviewer #3

(Remarks to the Author)

The manuscript by Kindlova et al provides an allele-resolved nanopore map of the human placental methylome. The paper is brief, well-written and easy to understand even though I must admit that most of the presented results do not come across as particularly surprising or novel. Nevertheless, the generated datasets have resource value and could be of use for researchers studying human developmental epigenetics. The major advance associated with this manuscript is likely the expansion of the placental DMR catalogue and the identification of novel imprinted genes, however those analyses appear incomplete and fall short of breaking novel ground. To give these results more credibility, I would urge the authors to conduct the following experiments / analyses:

- 1) Validation of DMRs by an orthogonal approach i.e bisulfite sequencing. Perhaps low coverage WGBS would be sufficient to validate differential DNA methylation across regions identified by Nanopore sequencing. RRBS could also be an option, if the identified regions are CpG-rich, which they likely are. Alternatively, comparisons with publicly available WGBS placental methylomes could be sufficient.
- 2) It would be very useful if the authors could use the wealth of already available data to provide more context related to the identified DMRs (both novel, and previously described). For example, analyses of available histone modification, ATAC-seq, ChIP-TF data, as well as sequence composition. Such analyses would surely boost the resource value of the paper.
- 3) In regard to novel imprinted genes, the authors discuss a few notable examples, yet it is not clear how many have been identified. Again, some validations here would be very welcome. For example, could the authors compare the DMRs associated with these novel imprinted genes to available WGBS data? Even though such publicly available data will not be allele-resolved, just looking at the reads covering DMR loci would already be very useful, as the expectation would be that ~50 % of the reads would be hyper- and 50% hypomethylated.

Version 1:

Reviewer comments:

Reviewer #1

(Remarks to the Author)

The authors have thoroughly engaged with the comments including generating new data, performing complementary analyses and improving the clarity of the figures. I recommend the manuscript for publication.

(Remarks on code availability)

Reviewer #2

(Remarks to the Author)

There are response comments in the rebuttal document that are not propagated to the manuscript or supplement (or that I'm having trouble finding where they are located based on the rebuttal text). It's common that manuscript readers would have similar questions to the reviewers, so this is curious.

An example is that the response document says that the 8 samples in this study were a random sub-sample of the QFC that had female babies and uncomplicated pregnancies. This information is highly relevant to interpreting the results, and I don't see it in the methods section of the manuscript text.

As a follow-up, were these placental tissues from cesarean or vaginal deliveries? What was the gestational age at delivery?

The code availability statement takes you to a github repository for methylartist, which is a method published by this team in 2022. It does not take you to the specific code to produce the results tables/figures for this paper. It is not possible to replicate this study without the specific code.

(Remarks on code availability)

The code availability statement takes you to a github repository for methylartist, which is a method published by this team in 2022.

It does not take you to the specific code to produce the results tables/figures for this paper.

It is not currently possible to replicate this study with the code provided.

Point-by-point response to reviews

Modifications to the manuscript in light of these reviews are highlighted in yellow in the manuscript.

Reviewer #1 (Remarks to the Author):

In this manuscript, Kindlova, Byrne et al. present a phased methylome analysis of human female placentae. The study design is excellent, with trio data and high levels of replication for this type of study (8). The underlying data (high-depth nanopore + Illumina) is excellent as well. The authors confirm the lower methylation levels in placenta relative to most other tissues, call allele-specific DMRs including some new ones (pending comparison with more recent studies), describe 2 new imprinted genes, and detect somatic mutations.

While the analyses are generally well done, the manuscript is mostly illustrative: the approach of using nanopore for phased methylomes is already well described and the analyses are not pushed far enough to test the hypotheses the authors put forward as explanations for the patterns they see. The impact for either the nanopore genomics or the epigenetics field is therefore limited.

Comments:

1. the manuscript should acknowledge at least two precedent studies for nanopore-based phased methylomes: Gigante et al. NAR 2019 (placenta imprinting already, but in mouse) and Akbari et al. eLife 2022 (imprinting in 12 human B-lymphocyte cell lines)

Thank you, these are now cited in the introduction at the end of the second paragraph.

2. The Akbari DMRs should be included in the cross-reference for known DMRs

While these are not DMRs in placental tissue, we agree there is value in cross-referencing the DMRs present in Supplemental Table 3 with the DMRs in Supplemental Table 4 from Akbari et al 2022. This has been done and overlaps are presented in Supplemental Table 3 in the column "Akbari2022".

3. what characterises the DMRs that were previously missed? Do they belong to certain regions that couldn't be phased before, or are they tissue-specific? Were there known DMRs that couldn't be found? Can the DMRs be classified into primary and secondary imprints?

Primarily it's the combination of limited ability to phase short reads coupled with most DMRs being tissue specific; see Figure 3b and discussion around this in the last paragraph of Page 4. There may also be a subset that are sex-specific which would be missed in our all-female cohort. Related to the short vs long reads factor, one might anticipate that there is a

difference in mappability between novel DMRs and those previously detected: the mean mappability as measured via BisMap 100bp is higher for previously detected DMRs versus novel ones (0.968 vs 0.876, $p=2.8e-10$ Welch two sample T-test), mappability has been included in Supplemental Table 3. We sought to be as inclusive as possible, so “known DMRs” were sourced from supplemental materials tables with varying false positive rates across several studies. While a good question, it isn’t clear how one could definitively separate secondary imprints, i.e. those occurring somatically, using this data, and we could consider this beyond the current scope of this study.

4. Please give more details about how the DMRs were selected. The methods imply that DSS was used to generate candidates, which are then selected visually?

That is correct. DSS was used to generate candidates by comparing maternal to paternal genomes, ranked by a statistic that aggregates DMLs into DMRs. Candidates were then screened visually using methylartist allele-specific locus plots +/- 20kbp around each candidate DMR.

5. Care should be taken with the terminology around “imprinted”. For gene expression, just say “paternally expressed” or “maternally expressed”. “Paternally imprinted” is ambiguous, usually it means it’s the paternal allele that carries the primary imprint, but it says nothing of whether the imprint silences or activates the gene.

We agree this clarification is helpful and have added terms like “demethylated” and “expressed” (e.g. “maternally demethylated and expressed imprinted gene”) to specify exactly what we mean where appropriate.

6. Easily testable hypotheses should be tested: 1) PMDs contributing to lower placenta methylation, why not segment the PMDs and quantify them? 2) clonal skew in X chromosome inactivation, why not compare the skews to the Xist promoter methylation skew?

1. We would respectfully argue that the contribution of PMDs to the overall lower genomic methylation level in placenta is visually and logically obvious. 2. We’ve done this and the X chromosome-wide differential methylation pattern with respect to parent-of-origin (Supplemental Figure 4) generally reflects the differential methylation of the XIST promoter (Supplemental Figure 5).

7. The read methylation heatmaps are far too cramped, I can’t tell where one sample begins and the next starts. Since the manuscript emphasises visualisation, these should be improved. The “expression space” transform is also not intuitive. Lining up this plot with the gene annotation misleads into comparing the two tracks, when it’s basically impossible to do so. Traditional read pile-ups over exons would be much easier to understand, or simply a dot plot of mat/pat cpms for each sample. The genomic position markers are also very hard to parse on all methylartist plots, so it’s hard to tell the scale of the plot. One could add commas (100,000,000 b) or simpler units (100 Mb).

These are not intended to be interpreted as heatmaps, rather, they are meant to convey an overall sense of how the methylated and unmethylated CpGs are positioned within read mappings. These are probably unnecessary if they complicate interpretation and so have been removed from primary figures. As with “CpG space”, the “expression space” transform is meant to improve interpretation of what is otherwise a very sparse read mapping space due to a paucity of RNA-seq reads containing informative heterozygous variants. As it is a novel visualisation we agree its interpretation may be more complicated than what is required to make our point about maternal vs paternal expression. We have therefore removed expression-space plots and added dot plots showing maternal and paternal expression levels for samples where informative variants were present (note this is not possible for all samples - if no phased heterozygous SNPs are present in the RNA-seq reads for a given gene the sample is skipped). Units have been made more readable as suggested.

8. NA19240 (Fig 3a) has an opposite bias, with more paternal methylation. Is this expected? Is this observed in other replicates (see Akbari)? For Fig3b clarify that the set of DMRs is the 723 DMRs identified in this study. For Figure 3b, use a divergent scale from -1 to +1 to show that the methylation difference in the other tissues follows the same direction (or not).

While a good suggestion, we do not have parental data for most samples in 3b so it isn't possible to determine direction with respect to maternal vs paternal methylation.

9. For Fig4a, colour by sample and add %variance explained to the axes.

We have added colours to distinguish samples and added the amount of variance explained by each PC to the axes, which we agree is an improvement.

10. Is 100 somatic mutations a lot? Give a quantitative comparison with previous estimates for placenta and other tissues.

As noted these results are well in line with Coorens et al 2021, and we would direct the reviewer to Coorens et al 2021 (doi: <https://doi.org/10.1038/s41586-021-03345-1>) for further discussion around somatic mutations in placenta relative to other tissues. We would caution against direct comparison between this and the Coorens et al study as ascertainment methods differ: we retain only concordant calls between ONT and Illumina data, consider indels in our count in addition to substitutions, and apply a statistical threshold for VAF < 0.5. Coorens et report an average of 145 substitutions (range of 38–259) per sample. With our VAF cutoff and considering only substitutions (Supplementary Table 5) we find an average of 95 substitutions with a range of 37 to 150. Without the VAF filter we find an average of 172 substitutions with a range of 94 to 230. Both of our ranges with and without the VAF filter are contained (or very nearly so) within the Coorens et al range.

11. give statistics about %genome or %data haplotyped.

The amount of data in terms of fraction of reads and fraction of bases haplotyped has been added to Supplemental Table 1.

12. There's an opportunity to comment in the discussion on how one would conduct this study today: with R10.4.1 chemistry, there would be no need for Illumina data, and allele-specific expression would be much better quantified with long-read transcriptomics (e.g. PCB111).

This is true, and we are not using Illumina data for our ongoing work in this area using the R10.4 chemistry. Long-read sequencing from placental tissue doesn't have quite the same advantage as it might in other tissues or in cell lines due to especially high RNase activity in placental tissue; the best RIN scores we typically see from placental tissue are around 7 (typically lower), meaning long-read cDNA methods which use poly-A selection would be expected to show excessive 3' bias. This is the rationale for our using ribosomal RNA depletion coupled to short-read sequencing rather than long-read methods for transcriptome analysis. That said we would be interested in exploring ribosomal depletion followed by long-read cDNA sequencing; while it might be degraded it's likely to be more informative than short-read for allele-specific expression.

Reviewer #2 (Remarks to the Author):

Summary:

The placenta is a developmentally critical and relatively understudied tissue, which is highlighted in the current paper. In a subsample of the Queensland Family Cohort, 8 families were selected for analysis. In each family, maternal and paternal PBMCs were whole genome sequenced using standard, Illumina short read approaches. Placental tissue DNA was measured Nanopore long read sequencing, which allows for both sequence variant calling, as well as modified cytosine calling for DNA methylation assessment. The maternal and paternal genomes were used to phase the placental genome and test for differentially methylated regions between the maternal and paternal genomes. They identified 723 DMRs, including 536 that were consistent with previous reports. They also measured placental gene expression and linked DMRs to differentially expressed genes. Two regions, related to ILDR2 and RASA1 were highlighted as novel and with interesting biology. Finally, they noted somatic mutation burden in the placenta. This manuscript represents a creative use of a cutting-edge technology to understand the placental omics landscape. It provides an important contribution to the field, and I am highly enthusiastic about future opportunities to investigate these newly characterized imprinting regions in other cohorts.

Critique:

The authors begin the introduction by mentioning the unique cohort of cell types in the placenta. Have the authors estimated the cell composition in their samples? Methods to estimate placental cell composition using sorted-cell DNA methylation (PMID: 33407091) and single cell RNA sequencing (PMID: 36914823) profiles are available. I'm

curious if the between sample variability in DMRs observed may be reflective of a differential abundance of a specific cell type that is driving the imprinting signature.

This is an excellent suggestion - we have carried out an analysis of cell type composition using CIBERSORTx with the data from Campbell et al. 2023 (PMID: 36914823) as a reference as suggested. This has been presented as Supplemental Figure 9; overall the cell type composition is not substantially different between samples. This is consistent with the multi-site sampling and homogenisation protocol (Global Pregnancy CoLab, Burton et al 2014) used by the Queensland Family Cohort team as has been further detailed in Methods.

While this is a very compelling scientific story, there are elements of the communication that could be adjusted to facilitate interpretation for the reader. For example, many of the figure axis and color bars lack labels, requiring the reader to sort through long legends and make assumptions. The black/white/blue/green fuzzy heatmap in figure 2 and 5 is particularly difficult to interpret as I am not seeing any closed dots at the current resolution, I don't know the meaning of all of the colors, and the number of rows appears to be more than the number of samples.

We've removed the read-level methylation plots from main figures (see Reviewer 1 item 7). Additionally, we have worked to address unlabeled axes where the units might not be obvious.

For future cohorts to compare with this work and assess generalizability, the Methods section would benefit from a section on participant recruitment and sample collection. It was part of a larger cohort, but how were these 8 families specifically selected? When was parental blood drawn (at delivery?)? How were placental tissue samples collected and archived? Was maternal blood washed from the placentas?

We selected 8 families from the Queensland Family Cohort with female babies and uncomplicated pregnancies aimed at an unbiased cross-section of the cohort. Regarding placenta processing, this followed the Global Pregnancy CoLab protocol (Burton et al 2014), which is now better documented in methods. Blood was washed away with sterile saline followed by multi-site sampling and homogenisation. Parental blood was drawn at multiple points as per the Queensland Family Cohort protocol, we used samples drawn at 24 weeks.

Methods: For the genetics data, how many base pairs were the long and short read sequences? How many million reads were generated per sample?

This is presented in Supplemental Table 1 for all data types in the study.

Methods: "Genome-wide phased methylation values at each CpG were output via "methylartist wgmeth --dss"... This statement would be useful to have a citation so others could find this method. What significance level and magnitude level were used to call a DMR?

Citations to methylartist and DSS are now included at that point in Methods.

Methods: Especially for your placental DNA methylation readers who more commonly use bisulfite sequencing and are newer to Nanopore, it may be useful to have a sentence in the methods that says how the DNA methylation levels were quantified in this paper.

With nanopore sequence data, methylation is detected directly via the basecalling model, in this case of this study: “Bases (A, C, G, T, 5mCpG) were called with Megalodon 2.5.0...” (methods). Quantifying methylation at a particular CpG is then simply a matter of counting 5mCpG vs CpG in the bases in the reads aligned over that site. Here, this is done via Methylartist, which we have clarified in the methods section.

While I am enthusiastic that the data will be available, I did not see a code availability statement.

A code availability statement has been added.

Discussion: Can the authors put these DNA methylation observations about X-inactivation and imprinting in the context of developmental timing, relative to their measurement in placental tissue at term? When during development does X-chromosome inactivation take place and is it reasonable to assume that the size placenta pieces measured in this paper all came from the same clonal expansion, or might there be more than one clone represented in these 35mg of tissue? Similarly, when during development do the two waves of epigenetic wiping take place and how does that relate to this term collection?

Our observations around X-inactivation are mostly technical in nature and likely do not contain information about early development. The tissue samples are a mixture of sites sampled in the placenta (information about placental sampling has been further detailed in the methods). Within a single clonal expansion, one might expect X-inactivation to have silenced a consistent X allele whereas with our samples there is a bias in some towards one X chromosome or the other, with the direction and magnitude of the bias likely depending on the mix of clones in the sample. The first reprogramming wave occurs early after fertilisation and the second occurs in primordial germ cells, long before placenta samples are collected at birth.

I think that Supplemental Table 3 is going to be incredibly useful for other cohorts to examine replication or enrichment of these imprinting sites. Could the authors expand on the column titles or include a table legend to explain all of the information that is included. For example, in the column “Placenta_Mean_DMe”, which parent is the reference group? It looks like maternal, since many of the values are negative, but want to be sure. Is there a p-value column? I really like how they point out DMRs that were highlighted in other papers in this table. Were there imprinting regions suggested in those other papers that were not identified here?

We have added a legend to Supplemental Table 3 (see “Legend” tab). We sought to be as inclusive as possible in determining whether DMRs had been noted previously; “known

DMRs” were sourced from supplemental materials tables from the cited studies with varying false positive rates dependent on the study.

Reviewer #3 (Remarks to the Author):

The manuscript by Kindlova et al provides an allele-resolved nanopore map of the human placental methylome. The paper is brief, well-written and easy to understand even though I must admit that most of the presented results do not come across as particularly surprising or novel. Nevertheless, the generated datasets have resource value and could be of use for researchers studying human developmental epigenetics. The major advance associated with this manuscript is likely the expansion of the placental DMR catalogue and the identification of novel imprinted genes, however those analyses appear incomplete and fall short of breaking novel ground. To give these results more credibility, I would urge the authors to conduct the following experiments / analyses:

1) Validation of DMRs by an orthogonal approach i.e bisulfite sequencing. Perhaps low coverage WGBS would be sufficient to validate differential DNA methylation across regions identified by Nanopore sequencing. RRBS could also be an option, if the identified regions are CpG-rich, which they likely are. Alternatively, comparisons with publicly available WGBS placental methylomes could be sufficient.

We have generated whole-genome methylation data using enzymatic methylation sequencing (EM-seq,(Vaisvila et al. 2021)) for two samples (054 and 103) as an orthogonal method, and have added functionality to our methylation analysis software to incorporate C-T substitution data, permitting direct comparison to our nanopore-derived methylation data (see changelog for methylartist version 1.3.0). On a genome-wide scale, Supplemental Figure 2 demonstrates that EM-seq and nanopore sequencing yield highly comparable methylation profiles (Pearson’s $r \sim 0.98$), as do locus-level methylation profiles. Assignment of short-read data to one haplotype or the other using heterozygous variants (“read-backed phasing”) yields very sparse resolution of methylation on either allele. Allele-specific methylation with the short read (Illumina) EM-seq data in both samples is possible at 155 DMRs, and is concordant with EM-seq in one or both samples at 146 DMRs (94%). We would caution against referring to this as “validation” as we feel nanopore sequencing is more likely to represent the true methylation status given the very direct nature of methylation inference via this method, coupled with long reads being much more readily phased into maternal or paternal in origin as compared with the short-read method. The EM-seq data for all DMRs here EM-seq had any allele-specific coverage are now included as Supplemental Table 4.

2) It would be very useful if the authors could use the wealth of already available data to provide more context related to the identified DMRs (both novel, and previously described). For example, analyses of available histone modification, ATAC-seq, CHIP-TF

data, as well as sequence composition. Such analyses would surely boost the resource value of the paper.

While there is a vast quantity of chromatin occupancy and DNA binding data available through ENCODE and other projects, the placenta is an under-represented tissue. Additional columns have been added to Supplemental Table 3: DNaseI hypersensitivity from Human Villous Mesenchymal Fibroblasts (HVMF) and Mappability of bisulfite treated 100mers (BisMap100).

3) In regard to novel imprinted genes, the authors discuss a few notable examples, yet it is not clear how many have been identified. Again, some validations here would be very welcome. For example, could the authors compare the DMRs associated with these novel imprinted genes to available WGBS data? Even though such publicly available data will not be allele-resolved, just looking at the reads covering DMR loci would already be very useful, as the expectation would be that ~50 % of the reads would be hyper- and 50% hypomethylated.

To give a relatively stringent number, we find combined allele-specific differential methylation and allele-specific differential expression for 26 genes associated with DMRs not reported elsewhere and not otherwise noted as being imprinted. There are other ways in which a putatively imprinted gene might be novel – for example the DMR might appear in the supplementary information of a paper but not have been associated with allelic differential expression of a particular gene.

As noted in our response to your query #1, in many cases, some level of allele-specific resolution is possible. In addition it is perhaps worth noting that each of the 8 samples in our cohort serves as a biological replicate, and we do see consistency across samples for the sites we claim are differentially methylated and differentially expressed between alleles.

Changes to the manuscript have been highlighted in yellow

Reviewer #1 (Remarks to the Author):

The authors have thoroughly engaged with the comments including generating new data, performing complementary analyses and improving the clarity of the figures. I recommend the manuscript for publication.

We appreciate the time and effort taken to provide suggestions, which has resulted in an improved manuscript.

Reviewer #2 (Remarks to the Author):

There are response comments in the rebuttal document that are not propagated to the manuscript or supplement (or that I'm having trouble finding where they are located based on the rebuttal text). It's common that manuscript readers would have similar questions to the reviewers, so this is curious.

An example is that the response document says that the 8 samples in this study were a random sub-sample of the QFC that had female babies and uncomplicated pregnancies. This information is highly relevant to interpreting the results, and I don't see it in the methods section of the manuscript text.

As a follow-up, were these placental tissues from cesarean or vaginal deliveries? What was the gestational age at delivery?

We apologise for our oversight in not including this in the first instance. This information and more are now included as Supplemental Table 7 and referred to in the Methods section of our manuscript.

The code availability statement takes you to a github repository for methylartist, which is a method published by this team in 2022. It does not take you to the specific code to produce the results tables/figures for this paper. It is not possible to replicate this study without the specific code.

Code and associated data for each figure are now included as Source Data (<https://doi.org/10.6084/m9.figshare.28636532.v1>) and in the following repository, cited in our manuscript: <https://github.com/adamewing/KindlovaByrne2025>. Additionally, bedMethyl files for both alleles of all placental samples are available at <https://doi.org/10.6084/m9.figshare.28593515.v1>.

Reviewer #2 (Remarks on code availability):

The code availability statement takes you to a github repository for methylartist, which is a method published by this team in 2022.

It does not take you to the specific code to produce the results tables/figures for this paper.

It is not currently possible to replicate this study with the code provided.

See response to review comments above for source data and code availability.